**RESEARCH**                                                                                               **Open Access**

# Deterministic transition of enterotypes shapes the infant gut microbiome at an early age

Liwen Xiao[1,2†], Jinfeng Wang[1*†], Jiayong Zheng[3†], Xiaoqing Li[3] and Fangqing Zhao[1,2*]

* Correspondence: wangjf@biols.ac.cn; zhfq@biols.ac.cn
†Liwen Xiao, Jinfeng Wang and Jiayong Zheng contributed equally to this work.
[1]Computational Genomics Laboratory, Beijing Institutes of Life Science, Chinese Academy of Sciences, Beijing 100101, China
Full list of author information is available at the end of the article

## Abstract

**Background:** The succession of the gut microbiota during the first few years plays a vital role in human development. We elucidate the characteristics and alternations of the infant gut microbiota to better understand the correlation between infant health and microbiota maturation.

**Results:** We collect 13,776 fecal samples or datasets from 1956 infants between 1 and 3 years of age, based on multi-population cohorts covering 17 countries. The characteristics of the gut microbiota are analyzed based on enterotype and an ecological model. Clinical information ($n = 2287$) is integrated to understand outcomes of different developmental patterns. Infants whose gut microbiota are dominated by Firmicutes and *Bifidobacterium* exhibit typical characteristics of early developmental stages, such as unstable community structure and low microbiome maturation, while those driven by *Bacteroides* and *Prevotella* are characterized by higher diversity and stronger connections in the gut microbial community. We further reveal a geography-related pattern in global populations. Through ecological modeling and functional analysis, we demonstrate that the transition of the gut microbiota from infants towards adults follows a deterministic pattern; as infants grow up, the dominance of Firmicutes and *Bifidobacterium* is replaced by that of *Bacteroides* and *Prevotella*, along with shifts in specific metabolic pathways.

**Conclusions:** By leveraging the extremely large datasets and enterotype-based microbiome analysis, we decipher the colonization and transition of the gut microbiota in infants from a new perspective. We further introduce an ecological model to estimate the tendency of enterotype transitions, and demonstrated that the transition of infant gut microbiota was deterministic and predictable.

## Background

Dynamics of the gut microbiota during early development not only has a considerable impact on childhood but also influences their health when children grow up [1, 2]. Unlike that in adults, the microbial community in infants presents a less complex and hypervariable pattern, especially during the first few years after birth [3]. The simple composition of the neonatal microbiota facilitated us to elucidate and highlight the

establishment of the symbiotic relationship between the host and microbes. In contrast, the high plasticity and dependence on external environments of the infant microbiota may provide an opportunity to use external intervention during the early stage to improve children's health in the future.

To date, most of the studies on the gut microbiota in early stages of life have examined cross-sectional samples collected from a single time point [4, 5], and the dynamics of gut microbiota from newborn to infancy is unclear due to the lack of high temporal resolution data. To overcome this shortcoming, several longitudinal cohorts with large sample sizes were recruited [3, 6]. According to a recent TEDDY study [3], the gut microbiota of infants may be roughly divided into three phases, namely developmental (3–14 months), transitional (15–30 months), and stable phases (≥ 31 months). However, the transition pattern and its driving forces during this process remain largely unknown. Recently, several studies have attempted to reveal the structure of infant microbiota using the concept of enterotype [7, 8]. Unlike the three familiar and consistent enterotypes (*Bacteroides*, *Prevotella*, and *Ruminococcus*) in adults, the infant enterotypes classified through each study were different. The difference may be related to the instability and high diversity of the microbiota from newborn to infancy, or it may result from different sampling stages or insufficient data size. In either case, uniform classification and systematic analysis of infant enterotypes are necessary to focus on the colonization and succession of the gut microbiota in early life.

In addition to the development involved in the natural growth cycle, some critical time points or clinical factors may play important roles in shaping the neonatal microbiota. It was reported that the relative abundance of Firmicutes, such as *Clostridium*, *Streptococcus*, and *Enterococcus*, was much higher in infants delivered by cesarean section. In contrast, *Bacteroides* and *Bifidobacterium* showed an evident enrichment in the development of infants delivered via the vaginal route [6]. Postnatal factors such as feeding habits and solid food intake are crucial as well [3, 9]. Several studies have reported differences in the gut microbiota of infants among different ethnicities or continents [5, 10], which implies that this diversity may derive from early life stages. The integrated analysis of these factors will help to better understand the early development of human microbiota.

Herein, we longitudinally collected fecal samples of neonates from China and further integrated over 10,000 metagenomic or 16S rRNA sequencing datasets of longitudinal fecal samples from 17 countries, spanning the first 3 years of life. Based on the largest population to date, the microbial assembly, succession, and maturation during early life were elucidated. Through the application of ecological models and the Markov chain, the transition rate of the gut microbiota in the early stages was quantified. Our findings provide comprehensive insights into the initial colonization and transition of the human microbiota through the analysis of enterotypes, which may increase our understanding of the microbiota dynamics in early life.

## Results

### Population characteristics

We enrolled a cohort of 101 Chinese full-term healthy neonates in this study; collected their fecal samples at birth, 6, 12, or 18 months of age; and then used metagenomic

shotgun sequencing to investigate their microbial community structures. We also retrieved metagenomic or 16S rRNA sequencing datasets of longitudinal fecal samples of newborns and infants from 19 public cohorts [6, 8–23]. In total, 13,776 fecal microbiome datasets of 1956 infants aged 1–36 months were obtained and analyzed after quality control and batch effect correction and included seven time points on average for each individual. This combined dataset covered seventeen countries of six continents, including Asia, South and North America, Europe, Australia, and Africa (Fig. 1A, Additional file 1: Table S1).

## Classification of the infant gut microbiota in early life based on enterotypes

Enterotypes were classified in the large population of this study (*n* = 13,776) according to previously described methods [24]. The microbial profiling resulted in four enterotypes (Fig. 1B, Additional file 1: Fig. S1 and Additional file 1: Table S2) with the bacterial genera belonging to phylum Firmicutes, *Bifidobacterium*, *Bacteroides*, and *Prevotella*, dominating enterotypes 1 to 4. These enterotypes clustered stably with

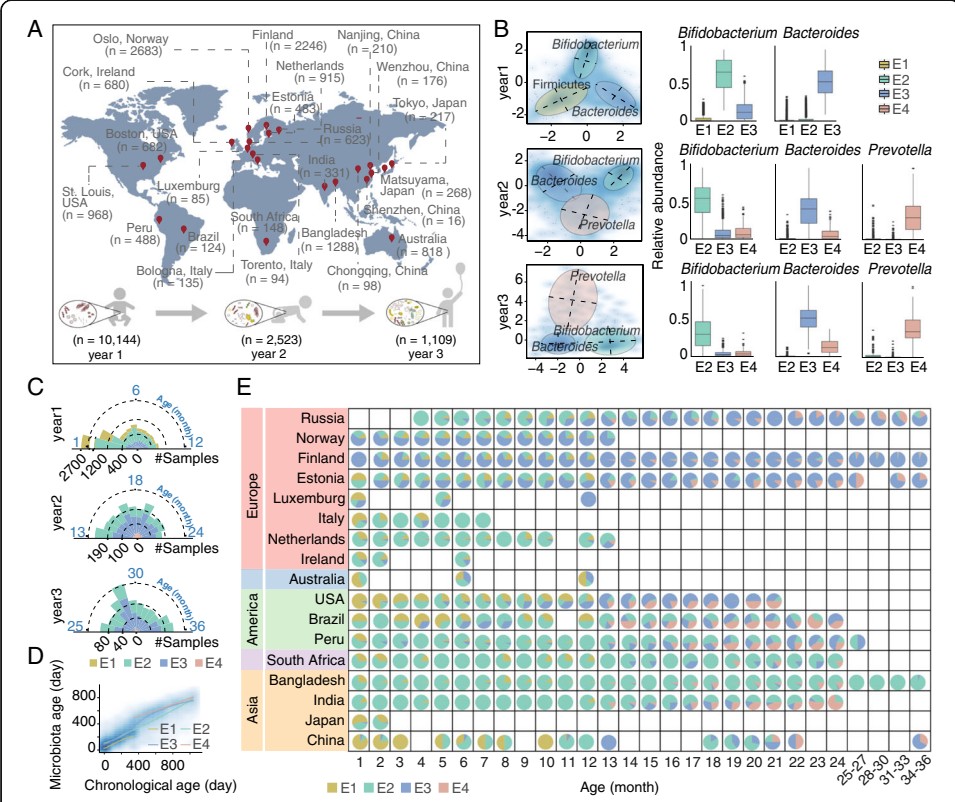

**Fig. 1** Characteristics of four infant enterotypes associated with temporal and geographic distribution. **A** Geographic (top) and temporal (bottom) prevalence of 13,776 stool samples used in this study. **B** Four enterotypes identified using Jensen–Shannon distance and partitioning around medoid (PAM) clustering in the first three years after birth. The colored ellipses cover 90% of the samples belonging to the enterotype group. The blue cloud (left) represents the local density estimated from the coordinates of stool samples. Box plots on the right show the relative abundance of the major bacterial contributor of each enterotype. **C** Prevalence of each enterotype in the first 3 years. **D** Longitudinal development of microbiome maturation based on the microbiota age throughout 3 years. The blue cloud represents the local density estimated from the coordinates of stool samples. **E** Temporal and geographic distribution of different enterotypes across 17 countries

various sample sizes as well as in different datasets (Additional file 1: Figs. S2-4), strongly indicating the reliability of enterotype clustering in infant gut microbiota.

To better understand the four enterotypes and to explore the temporal distributions of the infant microbiota in the first 3 years of life, we identified the emergence windows of each enterotype (Fig. 1B, C and Additional file 1: Fig. S5A-B). For enterotype 1 (E1, $n$ = 3062) dominant bacteria varied at the genus level (Additional file 1: Fig. S5A-B). In contrast, enterotypes 2 (E2, $n$ = 5264) and 3 (E3, $n$ = 5052), represented by a relatively high abundance of the genera *Bifidobacterium* and *Bacteroides*, respectively, were observed constantly in the first 3 years of life. Enterotype 4 (E4, $n$ = 398) dominated by the anaerobic *Prevotella* did not appear until the second year of life. Apart from the emergence windows, we also found that the community characteristics varied over time for each enterotype. Although the four enterotypes presented an increased pattern during early development both in diversity and microbiome maturation, E3 and E4 exhibited relatively higher alpha diversity and larger microbiome maturation rate than E1 and E2 after the first year of life (Fig. 1D and Additional file 1: Fig. S5C, Wilcoxon test, $P$ < 0.001), which might represent an advanced developmental stage of the gut microbiota in infants.

To investigate the geographical distribution of the infant microbiota in the first 3 years of life, we subsequently classified the four enterotypes in the populations of different countries. An evident geographical stratification of enterotypes was observed (chi-square test, $P$ < 0.001) (Fig. 1E and Additional file 1: Fig. S6). For example, among developing countries such as India, Bangladesh, South Africa, Peru, and Brazil, E2 was prevalent throughout the first 3 years of life. In contrast, developed countries in Northern Europe, such as Finland, Norway, and Estonia presented a *Bacteroides*-predominating E3 for most of the months. Besides these geographical differences, there was a clear trend of enterotype transition over time. For example, in Finland, the existence of E1 and E2 on the early stage was replaced rapidly by the large proportion of E3 with the growth of infants (Fig. 1E), which indicates a strong correlation between developmental stages and transition of enterotypes. Differences in the emergence windows and community characteristics of each enterotype as well as the enterotype preference in diverse countries suggest that the gut microbiota of these infants may be in distinct developmental stages, thereby representing different degrees of maturity.

### Distinct enterotypes correspond to different developmental stages of the infant gut microbiota

To further characterize the gut microbiota in the early stages of life along with the chronological ages and geographical distributions of infants, we compared the relative abundance of gut bacteria at the species or strain levels in four enterotypes based on 1165 metagenomic sequencing profiles from four countries (China, Luxemburg, Italy, and the USA).

As shown in Fig. 2A, the differences in gut microbiota were much greater among enterotypes than among countries, regardless of the infant age (ANOVA test, $R^2$ of enterotypes 4.65% vs $R^2$ of nations 2.88%, $P$ < 0.001). Specifically, no remarkable country-specific species were observed in this study, suggesting that the stratification of enterotypes may not directly result from geographical factors. In contrast, enterotype-

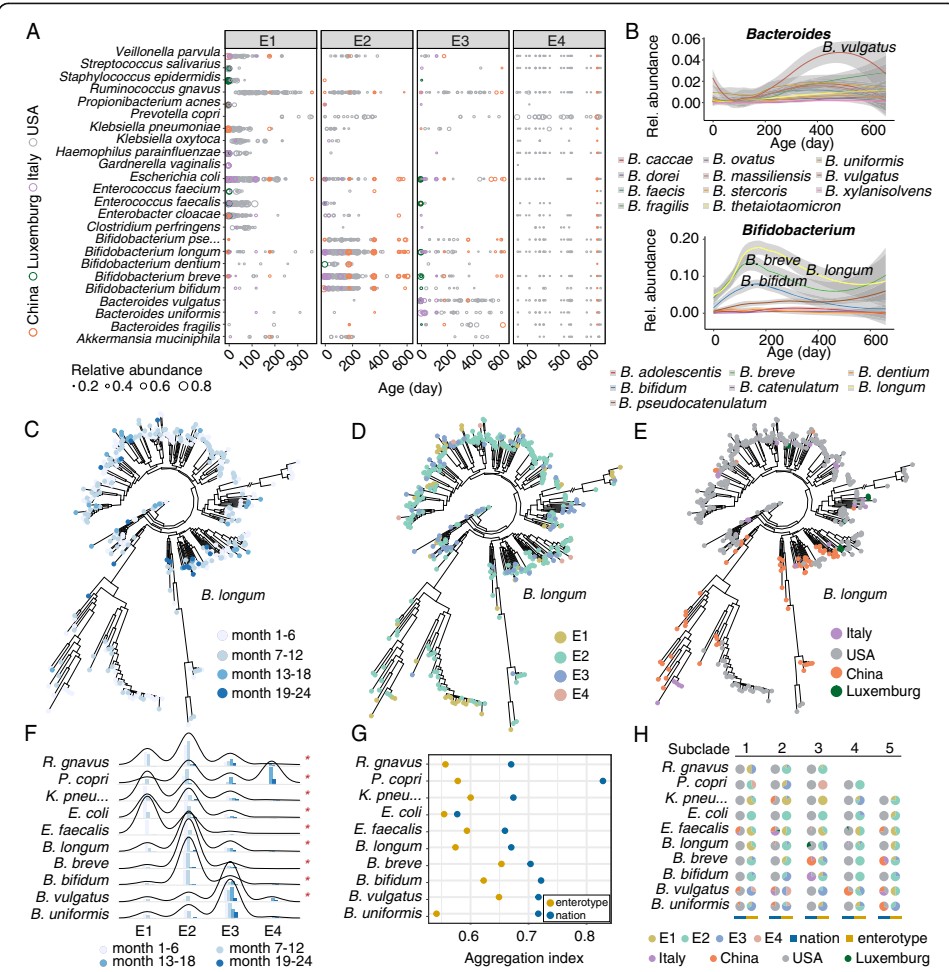

**Fig. 2** Species-level prevalence and strain-level association pattern of enterotypes. **A** Temporal distribution of the top 24 most dominant species among four enterotypes (n = 1165). The size of each circle is proportional to the relative abundance of each species. Only the species with relative abundance > 0.2 are shown. **B** The relative abundance of *Bacteroides* spp. and *Bifidobacterium* spp. varies over time. The shaded regions indicate the 95% confidence intervals for the fit of the lines. **C–E** Phylogenetic trees of different strains of *Bifidobacterium longum* denoted with developmental stages (**C**), enterotypes (**D**), and countries (**E**), respectively. Each node in the phylogenetic tree represents a specific strain from one infant. **F** Strain-level association between enterotypes and developmental stages in the top ten most dominant species. The height of each peak and bar indicates the number of strains enriched in this enterotype. Red asterisk on the right indicates a significant correlation (chi-square test, P < 0.001) between enterotypes and developmental stages. **G** Aggregation index (AI) quantifying the clustering closeness in the phylogenetic trees of top 10 species, where a higher AI value indicates a closer phylogenetic relationship among the strains in the same group (enterotype or country). **H** Pie diagrams of different enterotypes or countries in the top five subclades of representative species

specific species were found and were strongly associated with the main bacterial contributors of each enterotype, although the abundance of these species varied with the chronological age of the infants (Fig. 2B). This result indicates that even at the species level, distinct enterotypes may be clearly distinguished from each other and chronological ages may be an important factor associated with enterotypes in the development of infant gut microbiota.

To unveil the reasons underlying the differentiation of the gut microbiota in infants of different chronological ages and countries, we analyzed the microbiota at the strain

level. Unique marker genes were extracted from each sample and aligned to construct phylogenetic trees for the top ten most abundant species, and then subclades of each tree were counted (Additional file 1: Fig. S7). As an example, phylogenetic trees of *Bifidobacterium longum* for diverse developmental stages, enterotypes, and countries are shown (Fig. 2C–E). Notably, strains classified into E1 and E2 were precisely correlated to early developmental stages of the gut microbiota, while those from E3 and E4 were associated with later stages (Fig. 2C, D, chi-square test, $P < 0.001$). Regarding the geographic environment, the direct correlation was much weaker. Infants who belonged to the same developmental stages tended to share the same enterotype strain, regardless of the country of origin. For strains belonging to the same country, enterotypes varied in parallel with infant developmental stages (Fig. 2C–E). The strong correlation between enterotypes and developmental stages demonstrates that the age factor plays an important role on the stratification of enterotypes. The geographical stratification of enterotypes to some extent reflects the differences of developmental stages of infants in these countries.

To verify the correlation between enterotypes and the development of the gut microbiota, we further determined the enterotype association pattern in other species (Additional file 1: Fig. S7). Consistent with the observation in *Bifidobacterium longum*, the association between enterotypes and developmental stages was much stronger than that between enterotypes and geographical factors (Fig. 2F, G). Enterotypes dominated by *Bacteroides* and *Prevotella* exhibited a more mature pattern than the other two enterotypes, while strains from the same country were classified into different enterotypes due to the differences in the maturity of the gut microbiota (Fig. 2H), suggesting that the stratification of enterotypes among different countries can correspond to different developmental stages of the infant gut microbiota.

### The developmental process of infant gut microbiota is deterministic and predictable

To explore the developmental process of gut microbiota in the early stage, we stratified 1336 infants (only infants with more than three time points were included) into nine age groups with 4-month intervals to determine the enterotype transitions across different ages. As shown in Fig. 3A, a common inter-enterotype shift tendency was present throughout the entire period. The most frequent transition was observed in the first year of life. During this period, more than half of the infants (52.17%) in E1 shifted to E2, and some of the subjects previously belonging to E1 (9.24%) and E2 (9.12%) transitioned to E3. This transition, however, decelerated soon afterwards. In the second year, the microbiota became more stable, with over half of the infants (56%) without inter-enterotype variation, and in the third year, the ratio was up to 79.6%.

To quantify the transition of different enterotypes in early life microbiota, we used a Markov chain-based approach to model the enterotype transition probabilities (Fig. 3B). E1 showed a high frequency of transition to other enterotypes, with a relatively higher transition probability to E2 and E3 (0.37 and 0.39, respectively) compared to its self-transition probability (0.22). This may be the major reason for the disappearance of E1 in the following 2 years after birth (Fig. 3A). Further, E2 showed a tendency to transition to E3 (0.38). For this reason, although E2 corresponded to the majority of the infants in the first year of life, starting from the second year, E3 outnumbered E2 and

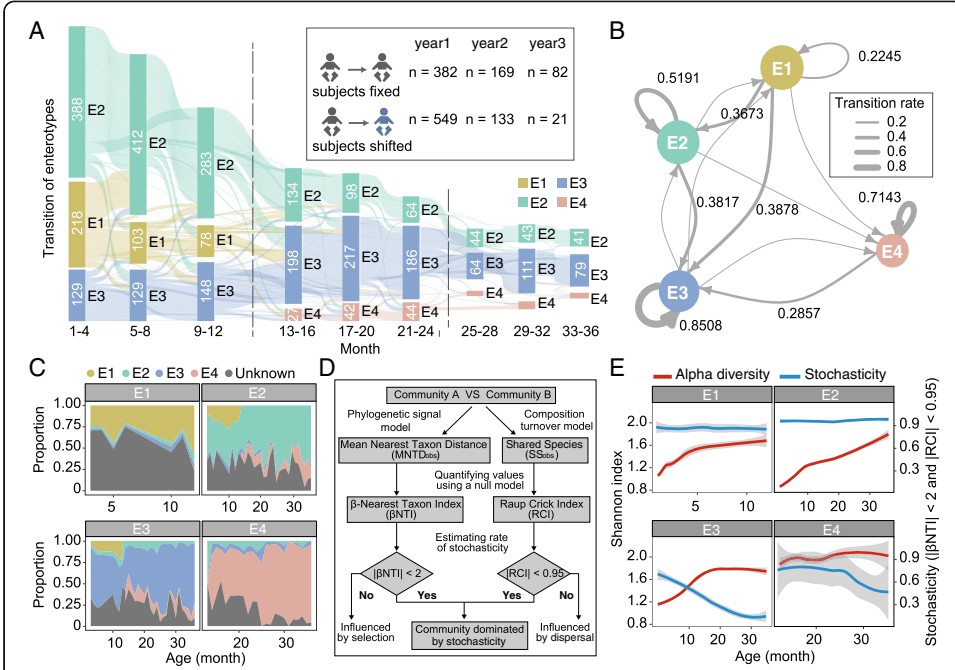

**Fig. 3** Temporal dynamics of four enterotypes in the first 3 years. **A** The Sanky diagram shows the transition of four enterotypes in the nine age groups ($n = 13{,}776$). The width of each flow is proportional to the number of infants experiencing an intra- or inter-enterotype transition. The height of each bar is proportional to the number of infants in each group. The number of intra- and inter-enterotype transition events in the first 3 years is summarized in the box. **B** Markov chain with subject-independent transition probabilities among four enterotypes, in which arrow weights are proportional to the maximum likelihood estimate of the transition probabilities among different states. Node size indicates the number of subjects in each enterotype. Only transition probabilities greater than 0.2 are shown. **C** Source tracking of infant gut bacteria in each time point. Samples from each enterotype of the former time point were considered as the potential sources of the latter time point. The areas of each color indicate distinct microbe source of each enterotype. The gray area indicates unknown sources. **D** A two-step procedure to evaluate the influence of stochasticity on the infant gut microbial community. β-nearest taxon index (βNTI) and Raup Crick index (RCI) were first calculated in each community. Then, the frequency of samples with |βNTI| < 2 and |RCI| < 0.95 in each month was used to determine the rate of stochasticity. **E** The temporal changes of alpha diversity and the rate of stochasticity in each enterotype. Red lines indicate alpha diversity based on the Shannon index and blue lines indicate the rate of stochasticity estimated from Fig. 4D. The shaded regions indicate the 95% confidence intervals

became dominant in infant populations (Fig. 3A). Changes in the abundance of their respective enterotype bacteria also accounted for the decline in E2 and the rise in E3 (Additional file 1: Fig. S5B). To estimate the transition and development of the four enterotypes in the future, we implemented a random forest model with a minimum gap from 30 to 60 days. It was observed that the transition of enterotypes in the early days of life was predictable with an AUC greater than 0.8, and the three enterotype bacteria (*Bacteroides*, *Bifidobacterium*, and *Prevotella*) played a crucial role in this prediction (Additional file 1: Fig. S8A). This result supported our assumption that the transition among enterotypes was a crucial path from stages of immaturity to maturity.

We further tracked the process of enterotype fluctuation by measuring the time-series changes of gut bacteria in infants, which helped us understand the alternation of microbes during the early development from a precise time scale. Each sample at the former time point served as the potential source to predict the origin of microbes at the latter time point in each enterotype. The results of source tracking exhibited

distinct patterns in each enterotype (Fig. 3C). We found a remarkably small proportion of microbes from the other three enterotypes imported into E1. In contrast, E3 accepted a large number of bacteria from E1 and E2. This tendency was consistent with the transition of enterotypes and strongly suggested that E1 might be the microbial source of the other enterotypes. Additionally, E3 contained the most microbes from the other enterotypes, thereby indicating its microbial sink status. Although we observed a limited frequency of microbial transmission from E4 to E2 and E3, a high prevalence of self-transmission of microbes in E4 (Fig. 3B, C) enhanced its stable feature compared to E1 and E2 and thus implied its maturation.

Since we have demonstrated that the less mature enterotypes (E1, E2) have an apparent tendency to transit to more mature enterotypes (E3, E4), a problem on the understanding of the reason underlying this transition awaits solution. We used an ecological model to explore the internal driving forces of each enterotype to elucidate potential factors affecting the microbiota dynamics (Fig. 4D). As shown in Fig. 3E, in the four enterotypes, the alpha diversity of the gut microbiota was positively correlated with the increasing age of infants, which indicated the gradual maturation of all enterotypes although the pace varied. However, the stochasticity ratios in E1 and E2 (~ 90%) were relatively constant and significantly higher than those in E3 and E4 (16–78%). Stochasticity ratios dropped sharply with increasing age in E3 (from 75.3 to 16.1%) and E4 (from 78.3 to 33.3%), which implied that these enterotypes were governed by the determined selection and presented a more stable community structure.

We subsequently measured the microbial interactions associated with the four enterotypes to explore the possible causations and trends of enterotype transition. A co-occurrence network at the species level revealed the difference in bacterial interactions (Additional file 1: Fig. S8B) and enhanced the credibility that the community structure of E3 and E4 was much more stable than that of E1 and E2. In these networks, both E1 and E2 presented a weak bacterial interaction, with most of their taxa separated from each other. In E3 and E4, however, the interaction was more frequent and the connection among species was much closer (chi-square test, $P < 0.001$). Particularly in E4, several species formed a compact *Prevotella*-centered cluster, which might contribute to the stability of their community structure and promote the maturation of the infant gut microbiota. Interestingly, we also found that the decrease of *Bifidobacterium* in E2 and the increase of *Bacteroides* in E3 were associated with the change in abundance of their corresponding bacteriophages (Additional file 1: Fig. S8C), suggesting that bacteriophages might be involved in regulating the abundance of enterotype bacteria during the transition of enterotypes.

### Multiple clinical factors are associated with the prevalence of enterotypes

To identify other factors that might affect microbiota dynamics, we analyzed external clinical factors obtained in this study (Additional file 1: Table S1). Intriguingly, the transition of enterotypes in early life was influenced neither by antepartum (maternal gestation age, the mode of delivery) nor by postpartum factors (the duration of breast-feeding) (Additional file 1: Table S3).

We then implemented PERMANOVA analysis ($n$ = 2287) and found that although multiple clinical factors could influence gut microbiota, the effects were much weaker

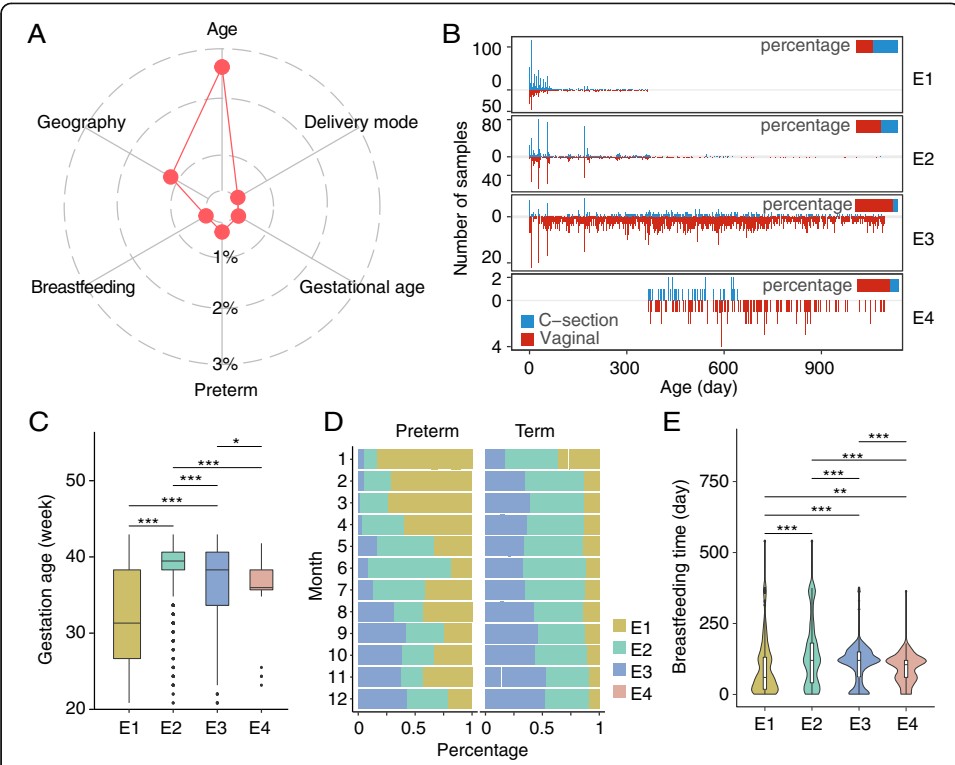

**Fig. 4** The association between four infant enterotypes and external factors. **A** The amount of variance ($R^2$) explained by multiple factors as analyzed using PERMANOVA (*adjusted P < 0.05, n = 2287*). **B** Prevalence of enterotypes associated with delivery mode (*n = 6494*). Red and blue lines indicate the number of vaginally and C-section delivered infants, respectively. Bar plots on the right indicate the percentage of infants in each enterotype associated with two delivery modes. **C** The boxplot of the gestational age of mothers in the four enterotypes (*n = 2986*). **D** The temporal frequency of the four enterotypes in full-term or preterm infants (*n = 10,144*). **E** Prevalence of enterotypes associated with breastfeeding (*n = 6139*). Wilcoxon rank-sum test, *\*P < 0.05, \*\*P < 0.01, \*\*\*P < 0.001*

compared with the infant age and the geographical factor (Fig. 4A), indicating that the bacterial community of infant gut experienced a rapid transition in the early age and the developmental stages played an essential role in this process. Considering individual differences may mask the feature of gut microbiota, we then explored the association between clinical factors and enterotypes separately. Among all the infants clustered into E1, over half were delivered by C-section (59.4%, hypergeometric test, $P < 0.001$). In contrast, in the remaining three enterotypes, children delivered via the vaginal route constituted a significant majority (hypergeometric test, $P < 0.001$) with 59.3% in E2, 89.6% in E3, and 78.9% in E4 (Fig. 4B). Such divergence was also observed at the genus level, in which the gut microbiota of C-section and vaginally delivered infants were dominated by the genera belonging to the phylum Firmicutes and genera *Bifidobacterium* or *Bacteroides*, respectively (Additional file 1: Fig. S9A). In addition, infants with E1 showed the lowest gestational age (Wilcoxon test, $P < 0.001$) (Fig. 4C). This tendency was more remarkable in preterm infants (Additional file 1: Fig. S9B), which clustered into E1 2–5 times more than those into E2 and E3 in the first 3 months (Fig. 4D). The correlation was also observed between enterotypes and the duration of breastfeeding; infants who were breastfed for a short (< 60 days) or long (> 300 days) term tended to be enriched in E1 and E2 (Fig. 4E). Despite the effects of clinical factors exerted on

gut microbiota were weak (Fig. 4A), there were associations between these factors and the prevalence of enterotypes.

**Divergence and transition of the metabolic capacity are associated with the development of infant gut microbiota**

To evaluate the potential roles of enterotype differentiation and transition on infant growth and development, we analyzed the functional variations of four enterotypes based on metagenomic data ($n$ = 1165). As expected, the four enterotypes grouped into distinct clusters at the functional gene level (Fig. 5A). Consistent with our previous analysis, E3 and E4 were markedly similar in functional gene profiles, although they possessed distinct dominant bacteria. E1 and E2 varied considerably, which suggested that these two enterotypes exhibited distinct functions although they both corresponded to the stage of the very early life.

Furthermore, the metabolic pathways showed marked enterotype- and age-specific patterns (Fig. 5B). A myriad metabolic pathway involved in arginine biosynthesis and branched amino acid biosynthesis were depleted in E1. In contrast, these amino acid biosynthesis pathways were enriched in all age groups of E2. The enriched metabolic pathways in E3 and E4 were similar, most of which were involved in glycolysis, starch degradation, and chorismate, phosphopantothenate, and queuosine biosynthesis.

We then discriminated bacterial contributors to the enriched pathways and found that the enriched amino acid biosynthesis in E2 was largely attributed to the genus *Bifidobacterium* (Additional file 1: Fig. S10). Additionally, we compared the bacterial taxa with the largest contributions to the functional features of the four enterotypes between the first 2 years (Fig. 5C). *Bifidobacterium longum* and *Bifidobacterium breve* accounted for the highest number of significantly enriched metabolic modules in the first year, whereas they were replaced by *Bacteroides vulgatus* in the second year. This was consistent with the trend observed in the enterotype transition that E2 was initially predominant in infants, but was later exceeded by E3 (Fig. 3A).

It has been demonstrated above that different enterotypes are related to different functions, and as infants grow, E1 and E2 shifted to E3 and E4. We next examined the longitudinal samples of each individual to investigate the association between enterotype transition and metabolic changes. We only chose infants with at least three sampling time points and excluded those without enterotype transition. Among the 257 infants with metagenomic sequencing samples, 119 were retained for downstream analyses. Nearly all these infants showed a dramatic synchronization between the transition of enterotypes and metabolic functions (*t*-test, *adjusted P* < 0.05) (Additional file 1: Fig. S11). For example, in infant TT0132A, the relative abundance of many amino acid biosynthesis pathways (L-arginine and L-isoleucine) and nucleotide biosynthesis (5-aminoimidazole ribonucleotide) decreased sharply when E2 shifted to E3 from day 133 to 205, while the relative abundance of E3-associated pathways (starch degradation and diacylglycerol biosynthesis) exhibited a gradual increase (Fig. 5D).

We finally divided infants who experienced stable transition ($n$ = 61) into two groups, group "E1/E2 to E3/E4" ($n$ = 53), which started with E1 or E2 and shifted to E3 or E4 at one stage of growth, and group "E3/E4 to E1/E2" ($n$ = 8), which showed an opposite direction in enterotype transition. Among the 32 significantly different pathways

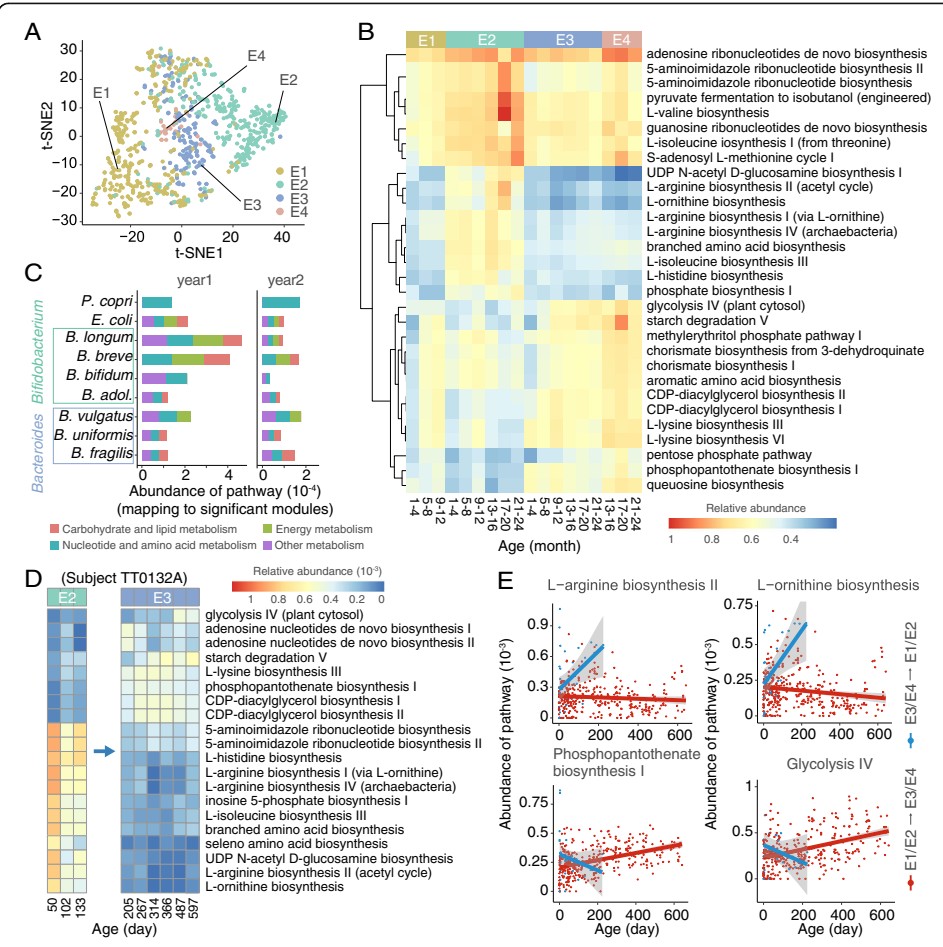

**Fig. 5** Divergence and transition of the metabolic capacity associated with the development of the infant gut microbiota. **A** t-SNE embedding based on Bray–Curtis dissimilarity matrices from the abundance of functional genes (*n* = 1165). **B** The thirty most significantly differentiated metabolic pathways of four enterotypes in the first 2 years of life. Pathways are hierarchically clustered according to their relative abundance. **C** Mean abundance of each significantly differentiated module binned at the pathway level. The nine representative species associated with four enterotypes are plotted. **D** The change of functional features associated with enterotype transition (from E2 to E3) in one American infant (subject TT0132A). The top 20 most differentiated pathways are presented. **E** An opposite trend over time was found in four representative metabolic pathways between immature (E1/E2) and mature (E3/E4) enterotypes. Infants have been stratified into two subgroups, in which red lines indicate infants with enterotype transition from E1/E2 to E3/E4, and blue lines indicate infants with enterotype transition from E3/E4 to E1/E2. Each dot represents the abundance of the respective pathway from one time point of one infant. The shaded regions indicate the 95% confidence intervals

between enterotypes (Fig. 5B), fourteen pathways diverged between these two groups (Additional file 1: Fig. S12). For example, L-ornithine biosynthesis and L-arginine biosynthesis showed decreased abundance in group "E1/E2 to E3/E4" with increasing age, but increased in the other group. An opposite trend was observed in the metabolic pathways related to phosphopantothenate biosynthesis and glycolysis (Fig. 5E and Additional file 1: Fig. S12). Collectively, these findings indicated that there was a strong correlation between the functional variation and the transition of enterotypes during the development of infants, and the shift from E1 and E2 to E3 and E4 was a common trend not only at the taxonomic but also at the functional level.

## Discussions

The dynamics of microbial communities in the infant gut have been highly discussed in recent years [2]. In this study, with over 10,000 longitudinal fecal samples of neonates spanning 17 countries, we analyzed the infant gut microbiota with the largest population to date and clustered four robust enterotypes for the first time. The results show that each of these enterotypes was extremely distinct from each other and was not only driven by different bacteria but was also associated with different phases of early development. In addition, a geography-related pattern of enterotypes was observed in the global populations. Our strain-level analysis further indicated that the stratification of enterotypes in different countries can be associated with different developmental stages of the infant gut microbiota. The explicit enterotype transitions and corresponding functional variations observed in this study illustrate the developmental trends of infant gut microbiota from the immature to the mature stages.

The enterotype concept has been debated since it was proposed in 2011 [24]. Many investigations have been performed on adult enterotypes [25, 26], while little has been researched on infants. A major critique raised by researchers is that enterotypes are not discrete states that separate individuals [27]. However, in this study, we observed that the enterotype clustering of infants was much more robust and consistent than that of adults, which might be due to the simplicity of the gut microbiota and limited influencing factors in early age. Apparent barriers among adult enterotypes, especially between *Bacteroides-* and *Prevotella*-dominated communities, were observed in previous studies [25, 28]. Unlike those of adults, infant enterotypes are vulnerable and tend to shift to another type. This inter-enterotype transition may result from the undergoing growth and physiological development of infants as well as external factors such as variable diets [5, 9]. This frequent transition stabilizes with age, thus denoting the maturation of gut microbiota from infants to adults. Batch effect is inevitable in meta-analyses due to different processing methods in different studies. In our study, however, no study-specific enterotype was observed during the clustering process, indicating that the enterotype clustering is independent of studies.

The TEDDY study divided the early development of infants into three distinct phases based on the diversity and richness of the gut microbiota. With over ten thousand fecal samples, they emphasized the effect of environmental factors on early development, especially the vital role of breast milk in this process [3]. The findings in our study are basically consistent with such a division that implies frequent fluctuations in the gut microbiota during the first few years of life, while the fluctuation recedes as neonates develop. In addition, the increasing metabolic capacity of amino acid and the importance of *Bacteroides* in the maturation of the infant gut are also in line with their observations. However, a novel finding in our study is that we observed a geography-related pattern in the stratification of enterotypes, as we have included multi-population cohorts covering 17 countries. Moreover, our study provides a new perspective on the transition of gut microbiota during the early development. This deterministic and predictable transition promotes gut microbiota from immaturity to maturity (Additional file 1: Fig. S13). The gut microbiota of infants in the TEDDY study was characterized with priority colonization of *Bifidobacterium*, while our study suggests that before the *Bifidobacterium*-dominant stage, there is an earlier Firmicutes-dominant stage, particularly in preterm infants, which is strongly associated with immaturity.

We further found that *Prevotella* was in relatively low abundance in the first few years of life and the *Prevotella* enterotype did not appear until the second year after birth. This finding was supported by a study that mentioned that the emergence of the *Prevotella* enterotype occurs much later than that of the *Bacteroides* enterotype [29]. The anaerobic characteristic and complicated carbohydrate-associated pathway of *Prevotella* indicate a more mature trait, which may account for the late appearance of this enterotype. Due to its low prevalence (13% in the second year and 6.2% the third year) in our study, the *Prevotella* enterotype was overshadowed by the other three more abundant enterotypes when using the full dataset, in which most samples that should have been classified into the *Prevotella* enterotype (72.9%) were falsely assigned to E3 (the *Bacteroides* enterotype). However, if we clustered enterotypes by sampling the same number of samples from each year, four distinct enterotypes were clearly observed, regardless of the sampling size selected. This clustering result is reproducible when using different clustering methods or even metagenomic datasets. These results emphasize the importance of sampling balance for enterotype clustering. Additionally, previous analyses of enterotypes among school-age children found that the *Bifidobacterium* enterotype showed the lowest gene number and diversity compared to the other two adult-like enterotypes [30], further demonstrating that the gradual decrease of the *Bifidobacterium* enterotype was correlated with the development of the infant gut microbiota.

Several studies have emphasized the importance of clinical factors like gestational period and breastfeeding option in early life [9, 31]. In this study, we confirmed that the maturity of gut microbiota in preterm infants was much delayed compared with that of full-term infants. Only 2287 samples were included in the multivariate analysis due to the lack of information from public studies. Although associations between clinical factors and the prevalence of enterotypes were observed in this study, their effects on gut microbiota were much weaker in the PERMANOVA analysis. We speculated that the divergence between two kinds of analyses may result from the variances of individuals because the geography and the infant age imposed great effects on gut microbiota, which may mask the influences of other factors. In addition, as most of preterm infants were delivered by C-section and fed with breastmilk, the associations we observed in the univariate analysis may be attributed to the influence of prematurity.

Another contribution of this study is that we introduced an ecological model to estimate the tendency of enterotype transitions. Some ecological concepts, such as the β-mean-nearest taxon distance (βMNTD) and Raup–Crick metric, have been widely applied in many natural ecosystems [32, 33]. We adapted this model to the gut microbial community and found that, in the two enterotypes (E1 and E2) in the early stages of life, the rate of stochasticity remained higher during the entire stage of life, while in the other two, it decreased sharply with the increasing age. Given that a high rate of stochasticity in a community indicates its disorder and instability [32], the modeling of stochasticity and selection on different enterotypes explain the trends of shifting from E1 and E2 towards E3 and E4 in the first 3 years of life. Functional features associated with this enterotype shift confirmed that as the infant developed, the four enterotypes displayed a determined metabolic transition, thereby altering the gut microbiota from the composition observed in an immature childhood to that observed in a mature adult-like stage. Nevertheless, research with higher resolution, such as a longer scale

span and more frequent sampling data, is required to understand the role of gut commensals in different stages of life as well as in different cohorts. Apart from this, the healthy status of the subjects should be associated with different growth patterns to determine outcomes of development in different communities in early life.

## Conclusions

In this study, we presented a comprehensive and quantitative enterotype analysis to elucidate the maturation of gut microbiota from infant to adult-like configuration using 13,776 longitudinal fecal samples from 1965 infants between 1 and 3 years of age, based on multi-population cohorts covering 17 countries. By leveraging the extremely large datasets and enterotype-based microbiome analysis in this study, we deciphered the colonization and transition of the gut microbiota in infants from a new perspective. The four enterotypes were correlated with different developmental stages of infants and exhibited obvious spatial and temporal patterns. We for the first time introduced an ecological model to estimate the tendency of enterotype transitions and demonstrated that the transition of infant gut microbiota was deterministic and predictable.

## Materials and methods

### Study cohorts and data retrieval

A Chinese cohort of newborns was recruited at the Wenzhou People's Hospital, and informed consent was obtained from their parents or guardians. The subject's mother was non-vegetarian and with no antibiotic use during pregnancy and had no history of smoking, alcohol consumption, or any other systemic or metabolic diseases. In total, 101 healthy children born at term were included. All children were of Han ethnicity, and their parents were permanent residents of Wenzhou city, China. Fecal samples were collected at birth, 6, 12, and 18 months of age. Stool was stored in study-provided sterile containers and kept at −20°C and transferred to −80°C upon return to the laboratory. DNA extraction from fecal samples was performed as per methods described previously [34]. The resulting DNA was stored at −80°C until sequencing. For each sample, a random-fragment library (insert length of ~300 bp) was constructed using the Nextera DNA Sample Preparation Kit (Illumina) with dual indexing and sequenced on the HiSeq 2500 platform (Illumina) to produce 150-bp paired-end reads. Sequencing generated an average of 84.4 million reads per sample, and 88.1% of the samples had >10 million reads. Initial FASTQ files were filtered prior to subsequent analysis using FASTQC. Additionally, sequencing data of 14,821 longitudinal fecal samples were retrieved from 19 public datasets [6, 8–23]. Of these datasets, three cohorts included metagenomic sequencing data and the others were 16S rRNA sequencing data; seven cohorts included preterm infants and twelve cohorts were infants born at term. Children with abnormalities related to growth (such as Down's syndrome, Turner syndrome, Fallot's tetralogy, multiple disabilities, and cystic fibrosis), and children treated with antibiotics during fecal sample collection were excluded. Finally, 1956 healthy children from 23 cities in 17 different countries were included in the study, and their fecal samples covered the first 3 years of life (1–36 months).

### 16S rRNA data processing

For 16S rRNA sequencing data profiling, raw reads were analyzed using the open-source software package QIIME [35]. This pipeline selected operational taxonomic units (OTUs) using a reference-based method and then created an OTU table. Briefly, high-quality 16S rRNA gene sequences were assigned to OTUs using the script pick_closed_reference_otus.py with a 97% identity threshold. OTUs were subsequently mapped to a subset of the Greengenes database [36]. Abundances were recovered by mapping the demultiplexed reads to the representative OTUs and by producing the final taxonomic profiles. Low-abundance OTUs, whose relative abundance did not reach 0.1% in at least 10% of the samples, were excluded.

### Metagenomic data processing

For metagenomic sequencing data profiling, human DNA sequences were identified and removed using KneadData (https://huttenhower.sph.harvard.edu/kneaddata/) by aligning raw reads to the hg19 human reference genome. The adapter and index sequences were trimmed and sequences were quality-filtered using Trimmomatic [37] with the following parameters: -jar trimmomatic-0.36.jar PE -phred33 ILLUMINACLIP: TruSeq3-PE.fa:2:30:10 LEADING:3 TRAILING:3 SLIDINGWINDOW:4:15 HEADCROP:8 MINLEN:36. The relative abundance of bacterial species was calculated using MetaPhlAn [38] with default parameters. Taxonomic tables with relative abundance were merged using the "merge_metaphlan_tables.py" script. The abundance of metabolic pathways was determined using HUMAnN2 [39]. Low-abundance filtering was applied to exclude taxonomic and functional features whose relative abundance did not reach 0.1% and 0.01%, respectively, in at least 10% of the samples.

### Enterotype clustering

The enterotype clustering was performed at the genus level according to the previous protocol [24]. The genus with the highest relative abundance in each year was considered as the main contributor of each enterotype. For E1, the main contributors were classified as phylum Firmicutes since the dominant genera possessed a similar relative abundance. To confirm clustering stability, we randomly selected different samples for enterotype clustering and reclassified enterotypes using each cohort. 16S rRNA gene sequencing data and metagenomic sequencing data were also used to reclassify enterotypes, respectively. We further clustered enterotypes by sampling the same number of samples from each year. Regardless of the sampling size selected, they were clearly clustered into four distinct enterotypes. Except for the partitioning around medoid (PAM) method, the Dirichlet multinomial mixtures (DMM) approach was also conducted to verify the clustering results [40].

### Microbiota maturation modeling

Random forest (RF) regression was performed to evaluate the microbiota age as previously described [41]. Briefly, the model was trained on 10% randomly selected full-term infants (> 37 weeks of gestation) belonging to each enterotype in the final dataset. To estimate the minimal number of the top ranked age-discriminatory taxa required for prediction, the rfcv function implemented in the "randomForest" package was applied

over 100 replicates. This model was then applied to all datasets, and the age of the infants predicted by this model was considered as microbiota age.

### Strain-level taxonomic and phylogenetic analysis

Bacterial strains that were present in multiple samples were identified using StrainPhlAn [42]. The sample-based strain reconstruction and reference databases of each clade and all the reconstructed genomes were analyzed to build multiple sequence alignments and phylogenetic trees.

We used an aggregation index (AI) to quantify the clustering effects through each variance. A cluster was defined as the largest subtree in each phylogeny with all the samples belonging to the same country or enterotype, which contained at least three samples. We defined the aggregation index of a country or enterotype for each species as follows:

$$\text{AI} = \left[\frac{\left(\sum N_\text{s}\right)^2}{\left(N_\text{a}\right)^2 \times N_\text{c}}\right]^{\frac{1}{10}}$$

where $N_\text{a}$ represents the number of all samples in each phylogenetic tree, $N_\text{c}$ the number of clusters in each phylogenetic tree, and $N_\text{s}$ the number of samples in each cluster. The AI values increased with the aggregation of clusters in each phylogenetic tree. All phylogenetic trees were visualized using the R package "ggtree".

### Tracking the dynamics and bacterial interactions of enterotypes

All samples were divided into nine groups with 4-month intervals, and the enterotype of each infant was considered to be the most frequent enterotype during each time scale. The inter-enterotype transition rate was quantified and visualized using a Markov chain based on a previously published R script [43]. Source tracking was applied using the R package "FEAST" [44]. To predict potential sources by month, we only focused on subjects with over three time points in our study. For each month, we randomly selected 100 samples that served as sources and sinks, and ten iterations were performed in this process to obtain average predictions.

### Evaluation on the stochasticity of the gut microbiota assembly

A two-step procedure was performed to estimate the rates of stochasticity in each community [33, 45]. First, the observed degree of phylogenetic turnover of each pairwise community comparison was quantified with the β-mean-nearest taxon distance (βMNTD) using the R function "comdistnt" (abundance weighted = TRUE; package "picante") [46, 47]. The βMNTD value quantifies the phylogenetic distance between each OTU in one community ($k$) and its closest relative in a second community ($m$):

$$\beta\text{MNTD} = 0.5\left[\sum_{i_k=1}^{n_k} f_{i_k}\ \min\left(\Delta i_k j_m\right) + \sum_{i_m=1}^{n_m} f_{i_m}\ \min\left(\Delta i_m j_k\right)\right]$$

where $f_{i_k}$ is the relative abundance of OTU$i$ in community $k$, $n_k$ is the number of OTUs in $k$, and $\min(\Delta i_k j_m)$ is the minimum phylogenetic distance between OTU$i$ in community $k$ and all OTUs$j$ in community $m$.

The degree to which βMNTD deviates from a null model expectation measures the degree to which the community composition is limited by selection on OTU ecological niches. The difference between the observed βMNTD and the mean of the null distribution was measured in units of standard deviation of the null distribution and is referred to as the β-nearest taxon index (βNTI):

$$\beta NTI = \frac{\beta MNTD_{obs} - \overline{\beta MNTD_{null}}}{sd(\beta MNTD_{null})}$$

where $\beta MNTD_{obs}$ is the observed βMNTD, $\beta MNTD_{null}$ is the null value of βMNTD, and sd indicates the standard deviation of the $\beta MNTD_{null}$ distribution. We quantified the βNTI for all pairwise comparisons using a separate null model for each comparison. βNTI values $< -2$ or $> +2$ indicate significantly less than or greater than the expected phylogenetic turnover, respectively. $|\beta NTI|$ values $< 2$ indicate the dominance of stochastic processes in communities [32, 48].

Second, a Raup–Crick metric was calculated to estimate the degree of turnover in OTU composition using R package "vegan" and R code of a previous study [49]. After modification, the Raup–Crick index (RCI) represents the dissimilarity between two communities relative to the null expectation. RCI values between $-0.95$ and $+0.95$ denote drift acting alone, which indicates stochastic processes exceeding determined processes in communities [49]. As a result, the rates of stochasticity were recognized from the proportions of community pairs that were between $|RCI| < 0.95$ and $|\beta NTI| < 2$.

### Bacteriophage identification and prediction model construction

We adopted a modified de novo CRISPR pipeline used in a previous study to identify bacteriophages [50, 51]. Thirteen enterotype-associated bacteria were first chosen to construct a direct repeat (DR) database. Reference genomes of these bacteria were downloaded from the NCBI database. DRs were identified from bacterial genomes using Piler with default settings [52]. We then extracted the interspaces from our metagenomic reads as CRISPR spacers using CRASS [53]. After comparing with contigs via BLASTn (mismatch $\leq 1$, and *E*-value $\leq 10^{-5}$), the target spacers were selected. DRs from the same region were compared with the DR database using BLASTn (*E*-value $\leq 10^{-10}$ and identity $= 100\%$). Significant hits were inferred as the phage source of the spacer (protospacer).

The RF package in R was used to predict the transition of enterotypes. All subjects sampled less than three time points were discarded. The remaining subjects were randomly divided into an independent training set (90%) and a testing set (10%). After tenfold cross-validation, 27 most important features were selected to predict the variation. We applied this model twice with a minimum gap of 30 and 60 days, respectively.

### Functional analysis

Metagenomic data were functionally profiled using HUMAnN2 [39]. Gene family abundance at the community level was calculated to show the contributions of known and unknown bacterial species. The pathway abundance was computed both at the community level and for each species using community- and species-level gene abundances along with the structure of the pathway. Functional clustering was performed based on

Bray–Curtis dissimilarity matrices of significantly different genes identified through Kruskal–Wallis tests.

## Statistical analysis

A traditional batch-correction method [54] was applied to remove batch effects. The variance explained by the enterotype was much larger than that of the study both before and after correction. All statistical analyses were conducted in R within RStudio and visualized using package "ggplot2." Because of the lack of sufficient metadata, all samples with clinical information were divided into four discrete groups (1, 6, 12, 18 months), and the quantification of the variance was calculated using PERMANOVA as implemented by the "adonis" function in the R package "vegan." Enterotype characteristics and variations associated with clinical factors were compared using chi-square tests for categorical variables and Wilcoxon rank-sum tests for continuous variables. For simple, independent comparisons, $P$-values $< 0.05$ were considered significant. For all analyses regarding multiple comparisons, we used the Benjamini–Hochberg method to correct for multiple testing.

## Supplementary Information

---

**Additional file 1: Tables S1-S3, Figures S1-S13.** Supplementary tables and figures.

**Additional file 2.** Reference List of Datasets. Reference list of datasets included in this study.

**Additional file 3.** Review history

---

### Acknowledgements
We thank all the participants involved in this study.

### Peer review information

### Review history
The review history is available as Additional file 3.

### Authors' contributions
FZ and JW conceived and designed the study and interpreted the data. JZ and XL collected the samples and conducted the experiments. LX, JW, and FZ analyzed the data and created graphs. LX, JW, and FZ wrote the paper. All authors approved the final version of the manuscript.

### Funding
This work was supported by the National Natural Science Foundation of China (32025009, 31722031, 31670119, 31870107, and 32070122) and the Beijing Natural Science Foundation (JQ18020).

### Availability of data and materials
Raw sequencing data have been deposited in the Sequence Read Archive database under accession number PRJNA695070 [55]. R scripts and metadata used for data analyses can be accessed at https://zenodo.org/record/5141515 [56]. Additional file 2 contains a list of previously published datasets used in this study.

## Declarations

### Ethics approval and consent to participate
The study protocol was approved by the medical ethics committee of Wenzhou People's Hospital and in accordance with the Helsinki Declaration. Informed consent was obtained from all participants (or the infants' parents).

### Consent for publication
Not applicable.

### Competing interests
The authors declare that they have no competing interests.

## Author details

¹Computational Genomics Laboratory, Beijing Institutes of Life Science, Chinese Academy of Sciences, Beijing 100101, China. ²University of Chinese Academy of Sciences, Beijing, China. ³Department of Gynecology and Obstetrics, Wenzhou People's Hospital, Wenzhou 325000, China.

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

## 
