## [**Additional file 3.** Review history · Genome Biology]

Review History

First round of review

Reviewer 1

Are you able to assess all statistics in the manuscript, including the appropriateness of statistical tests used? Yes, and I have assessed the statistics in my report.

Comments to author:

- Are the methods appropriate to the aims of the study, are they well described, and are necessary controls included? If not, please specify what is required.

Yes.

- Are the conclusions adequately supported by the data shown? If not, please explain

No

The authors report the large dataset (10k samples) as having 4 "enterotypes", however, the results displayed in Figure S1 clearly indicate that the data best fit to 3 enterotypes. It appears the Prevotella enterotype is not supported by the de novo clustering results of the full dataset. I understand that when the data is split up by year, then 3 clusters are expected (Figure S2) but the fact that the full dataset does not support the 4-cluster model is troubling. Unfortunately, this inconsistency leads to all following analyses being inappropriate. Perhaps this was an error in the choice of figure submitted, but I must take the data as presented, and this is a serious problem.

It is not clear to me that mixing 16S and metagenomic data at the genus level is valid, as the both methods have contrasting biases. This combination needs to be supported by analysis or precedent from another publication that examined the effect of combining these data types. For example, is there a confound between the enterotypes and the proportion of metagenomic data in each? It should be made explicitly clear how many samples of each 16S and metagenomic are in the data set (e.g. in Table 1).

The abstract and elsewhere are written in a misleading way. Some of their findings are based on over 10,000 samples, but much of the work is based on 1165 metagenomic datasets or 1336 samples from longitudinal data. It is not clear in every case (this should also be fixed), but it looks like on the results in Figure 1 use the large dataset.

This study design does not support making a recommendation for the length of time to breastfeed (in the discussion).

Are any of the metadata variables confounded? This should be tested and reported and, if necessary, considered in other analyses.

- Are sufficient details provided to allow replication and comparison with related analyses that

may have been performed? If not, please specify what is required.

No

The metadata and genus level profiles are not shared. This precludes any replication. Also, given that the authors are using the shared metadata from many other studies, it is unreasonable that they do not share their own metadata from the new samples included in this study so that other researchers may make use of their data, as the authors here have made use of the data from others. The curated metadata of all the samples used, as well as the genus and functional profiles, need to be shared to enable replication of the analysis.

- Does the work represent a significant advance over previously published studies?

The analyses and figures presented here show a large body of work and are generally well presented. The work presented is interesting.

However, the temporal development, major taxa, and transitions patterns were mapped in the TEDDY study. Having multiple voices in this space is valuable, so I do not think this should preclude publication, but an in-depth comparison to the previous work is required for this to be helpful to the readers and other researchers. For example, if the major difference is *Prevotella* abundance, do the authors have a hypothesis as to why there is this difference between the two studies?

- Is the paper of broad interest to others in the field, or of outstanding interest to a broad audience of biologists?

This paper would be of interest to those studying human microbiome development, the infant microbiome, and may be of interest to general microbiome audience.

Reviewer 2

Are you able to assess all statistics in the manuscript, including the appropriateness of statistical tests used? Yes, and I have assessed the statistics in my report.

Comments to author:

Xiao et al. characterized four enterotypes in the infant microbiome and their transition during the first 3 years of life, by making use of 13,776 fecal samples from 1,956 infants from 17 countries. This study has convincingly shown two less matured enterotypes (E1, E2) and two more mature enterotypes (E3 and E4), which were reflected by the differences of the diversity, bacterial co-abundance relationship and metabolic capacity between different enterotypes. Further associations also showed that the prevalence of enterotypes are linked to clinical factors. The authors also showed that the less matured enterotypes was more likely to transmit to matured enterotypes. However, this transmission was not associated with clinical factors. The paper is well written. Overall, the study is of great interest to the field.

Specific comments:

1. It is remarkable that the enterotypes were consistently observed in different infant cohorts across 17 countries. Some samples were sequenced by 16S while others were sequenced using shotgun metagenomics sequencing. However, we also know that different DNA isolation methods can generate very different microbial profiles. Did the authors check the DNA isolation methods used in different datasets? It would be important to assess their impact on enterotypes.
2. On page 6, the authors concluded that it is the developmental stage rather than the geographic environment that leads to the stratification of enterotypes. However, the figure 1E also shows that E3 were much more prevalent in some European countries (Norway, Finland and Estonia) but E2 were more prevalent in other countries at the early life. At late stage, E2 remained dominant in some developing countries (South Africa, Bangladesh and India). This statement needs to be clarified.
3. On page 9 and Figure 4, the prevalence of enterotypes were associated with various clinical factors (C-section, gestational age, preterm and breastfeeding). As the prevalence of enterotypes can differ at different age, it is unclear at which age these analyses were conducted? Do those factors have short-term or long-term impact on the gut microbiome?
4. On page 10, did the pathway analyses correct for multiple testing?
5. Figure S9 needs to add individual data points.

Authors Response

Point-by-point responses to the reviewers' comments:

Reviewer #1 (Remarks to the Author):

- Are the methods appropriate to the aims of the study, are they well described, and are necessary controls included? If not, please specify what is required.

Yes.

- Are the conclusions adequately supported by the data shown? If not, please explain

No

The authors report the large dataset (10k samples) as having 4 "enterotypes", however, the results displayed in Figure S1 clearly indicate that the data best fit to 3 enterotypes. It appears the *Prevotella* enterotype is not supported by the de novo clustering results of the full dataset. I understand that when the data is split up by year, then 3 clusters are expected (Figure S2) but the fact that the full dataset does not support the 4-cluster model is troubling. Unfortunately, this inconsistency leads to all following analyses being inappropriate. Perhaps this was an error in the choice of figure submitted, but I must take the data as presented, and this is a serious problem.

Response: We thank the reviewer's comments. The reason why the *Prevotella* enterotype did not appear in the clustering results of the full dataset is that a large majority of samples (73.6%) included in our study were collected from the first year and the abundance of *Prevotella* in the first

year of life is quite low. If the enterotypes were clustered using the full dataset of all 3 years, the *Prevotella* enterotype would be overshadowed by the other 3 more abundant enterotypes. However, if we split the samples into different years and clustered them separately, we can clearly see the *Prevotella* enterotype from the second year (**Figure 1B**).

To address the reviewer's concern, we further clustered enterotypes by sampling the same number of samples from each year. As shown in **Supplementary Figure 2A**, regardless of the sampling size selected, they were clearly clustered into four distinct enterotypes. This clustering result is reproducible when using different clustering method (e.g. Dirichlet multinomial mixtures) (**Supplementary Figure 2B**) or even metagenomic datasets (**Supplementary Figure 3**).

It is not clear to me that mixing 16S and metagenomic data at the genus level is valid, as the both methods have contrasting biases. This combination needs to be supported by analysis or precedent from another publication that examined the effect of combining these data types. For example, is there a confound between the enterotypes and the proportion of metagenomic data in each? It should be made explicitly clear how many samples of each 16S and metagenomic are in the data set (e.g. in Table 1).

Response: We have considered the potential biases caused by the mixing of 16S and metagenomic data in our study and re-clustered enterotypes using 16S and metagenomic data, respectively. The high consistency between the results of the separated data and the full data suggested that our enterotype clustering is robust and convincing (**Supplementary Figure 3**).

The numbers of 16S and metagenomic samples are shown in **Supplementary Figure 3A-B**.

The abstract and elsewhere are written in a misleading way. Some of their findings are based on over 10,000 samples, but much of the work is based on 1165 metagenomic datasets or 1336 samples from longitudinal data. It is not clear in every case (this should also be fixed), but it looks like on the results in Figure 1 use the large dataset.

Response: We thank the reviewer for pointing this out. Only the strain level (**Figure 2**) and functional analyses (**Figure 5**) were based on the 1,165 metagenomic dataset. All the other analyses were based on the full dataset (n = 13,776). The number shown in the box of **Figure 3A** represents the number of infants (1,336) but not the number of samples, as each infant has multiple longitudinal fecal samples. In this part, over 10,000 samples were included. We have clarified these points in the revised manuscript (Page 4 Line 26, Page 7 Line 12, and Page 10, Line 4).

This study design does not support making a recommendation for the length of time to breastfeed (in the discussion).

Response: We agree with the reviewer's comment, and as suggested, we have removed this part from discussion (on page 12).

Are any of the metadata variables confounded? This should be tested and reported and, if necessary, considered in other analyses.

Response: We thank the reviewer for this suggestion. As many public datasets used in our study did not provide detailed clinical information, for example, some only provided information on breastfeeding or the mode of delivery while others did not provide anything. Therefore, we only performed data analyses on those datasets with available clinical information, which were shown in **Supplementary Table 3**. These clinical factors did not affect the dynamic transition of

enterotypes during development. The multivariate analysis in **Supplementary Figure 8C** showed the effect of these factors weakened with the growth of infants.

- Are sufficient details provided to allow replication and comparison with related analyses that may have been performed? If not, please specify what is required.

No

The metadata and genus level profiles are not shared. This precludes any replication. Also, given that the authors are using the shared metadata from many other studies, it is unreasonable that they do not share their own metadata from the new samples included in this study so that other researchers may make use of their data, as the authors here have made use of the data from others. The curated metadata of all the samples used, as well as the genus and functional profiles, need to be shared to enable replication of the analysis.

Response: As suggested, the metadata and genus level profiles, including our code scripts for data analyses, have been uploaded and can be accessed at <https://github.com/xiaolw95/enterotype-analyses>.

- Does the work represent a significant advance over previously published studies?

The analyses and figures presented here show a large body of work and are generally well presented. The work presented is interesting.

Response: We greatly appreciate the reviewer's comments on the novelty and significance of our study.

However, the temporal development, major taxa, and transitions patterns were mapped in the TEDDY study. Having multiple voices in this space is valuable, so I do not think this should preclude publication, but an in-depth comparison to the previous work is required for this to be helpful to the readers and other researchers. For example, if the major difference is *Prevotella* abundance, do the authors have a hypothesis as to why there is this difference between the two studies?

Response: We greatly appreciate the reviewer's point of view that there should be a comparison to the TEDDY study. Actually, we did try to ask the authors to access their raw data from the TEDDY project at the beginning of our research. Unfortunately, we did not receive any responses and thus their data were not included in our analyses.

Nevertheless, we still compared our results with the findings in the TEDDY study. First of all, there is a strong consistency on the dynamics of infant gut microbiota between the two studies. For example, the abundance of distinct species of *Bifidobacterium* varied and the metabolic capacity of amino acids increased with infant growth. In addition, the genus *Bacteroides* was correlated to the maturity of infant gut. However, the major difference between our research and theirs is that they roughly divided the early development of infant gut microbiota into three phases but did not go deeper into the topic, and they focused on the effects of various factors on the development of infants, emphasizing the protective effect of breastfeeding. Our study provides a new perspective on the transition of gut microbiota during the early development. Compared to the TEDDY study, the main novel findings of this study are as follows:

- (1) We analyzed the infant gut microbiota with the largest population to date and clustered them into four robust enterotypes. Multiple clustering methods and sub-sampling

approaches were employed to demonstrate the high reliability of enterotype clustering in this study.

- (2) We found that the four enterotypes were strongly associated with the developmental status of infants, and the maturity of the infant gut microbiota was positively correlated with the development of the countries.
- (3) We elucidated the dynamics of infant microbiota and proposed that unlike to those of adults, infant enterotypes were much more vulnerable and exhibited a dramatic transition. We for the first time introduced an ecological model to estimate the tendency of enterotype transitions, and demonstrated that the transition of infant gut microbiota was deterministic and predictable.

- Is the paper of broad interest to others in the field, or of outstanding interest to a broad audience of biologists?

This paper would be of interest to those studying human microbiome development, the infant microbiome, and may be of interest to general microbiome audience.

Response: We greatly appreciate the reviewer's comments on the novelty and significance of our study.

Reviewer #2 (Remarks to the Author):

Xiao et al. characterized four enterotypes in the infant microbiome and their transition during the first 3 years of life, by making use of 13,776 fecal samples from 1,956 infants from 17 countries. This study has convincingly shown two less matured enterotypes (E1, E2) and two more mature enterotypes (E3 and E4), which were reflected by the differences of the diversity, bacterial co-abundance relationship and metabolic capacity between different enterotypes. Further associations also showed that the prevalence of enterotypes are linked to clinical factors. The authors also showed that the less matured enterotypes was more likely to transmit to matured enterotypes. However, this transmission was not associated with clinical factors. The paper is well written. Overall, the study is of great interest to the field.

Response: We greatly appreciate the reviewer's comments on the novelty and significance of our study.

Specific

comments:

1. It is remarkable that the enterotypes were consistently observed in different infant cohorts across 17 countries. Some samples were sequenced by 16S while others were sequenced using shotgun metagenomics sequencing. However, we also know that different DNA isolation methods can generate very different microbial profiles. Did the authors check the DNA isolation methods used in different datasets? It would be important to assess their impact on enterotypes.

Response: We greatly appreciate the reviewer's comments to improve our study. To test the probable biases caused by the mixing of 16S rRNA and metagenomic data, we re-clustered enterotypes using 16S rRNA and metagenomic data, respectively. The high consistency between the results of separated data and full data suggested that our enterotype clustering is robust and convincing (**Supplementary Figure 3**).

2. On page 6, the authors concluded that it is the developmental stage rather than the geographic environment that leads to the stratification of enterotypes. However, the figure 1E also shows that E3 were much more prevalent in some European countries (Norway, Finland and Estonia) but E2 were more prevalent in other countries at the early life. At late stage, E2 remained dominant in some developing countries (South Africa, Bangladesh and India). This statement needs to be clarified.

Response: We agree with the reviewer's comments that the correlation between enterotypes and development stages needs to be further clarified. The impact of developmental stages on enterotypes does not conflict with the geographical stratification of enterotypes. On the contrary, they may have a cause-and-effect relationship, in which the divergence of infant maturation in the developmental stages may lead to the stratification of enterotypes among different countries. For example, the development of gut microbiota of infants who lived in South Africa, Bangladesh and India may be delayed due to malnutrition or poor medical conditions. For this reason, the majority of infants in these developing countries remain in E2 and have not transitioned to E3, while infants living in some European countries mature faster with an overrepresentation of E3.

Even in these European countries (Norway, Finland and Estonia), there is a clear trend of enterotype transition over time. For example, in Finland, the existence of E1 and E2 on the early stage is replaced rapidly by the large proportion of E3 with the growth of infants (**Figure 1E**), which indicates a strong correlation between developmental stages and transition of enterotypes. The analyses at the strain-level further demonstrate that the developmental stages played an important role on the stratification of enterotypes (**Figure 2C-E**, Chi-square test, $P < 0.001$). In summary, the geographical stratification of enterotypes to some extent reflects the differences of developmental stages of infants in these countries.

3. On page 9 and Figure 4, the prevalence of enterotypes were associated with various clinical factors (C-section, gestational age, preterm and breastfeeding). As the prevalence of enterotypes can differ at different age, it is unclear at which age these analyses were conducted? Do those factors have short-term or long-term impact on the gut microbiome?

Response: We appreciate the reviewer's comments. All longitudinal samples of infants with clinical information were included in the clinical analyses (as most preterm infants in our study were from the first year, only the prevalence of the first year was shown in Figure 4C). The number of samples corresponding to each clinical factor was shown in **Supplementary Table 1**. To avoid possible biases, hypergeometric test has been conducted in these analyses. We have revised the manuscript to make it clearer (Page 9, Line 13).

Here we want to clarify that it is necessary to include all ages of infants when analyzing the impact of clinical factors on the prevalence of enterotypes. For example, we collected 36 longitudinal fecal samples during the first 3 years (one sample per month) from a vaginally delivered infant. If most of these samples were enriched in E3, we can say that this infant showed a tendency of transiting to E3; while if all vaginally delivered infants were enriched in E3, it may indicate that infants delivered via the vaginal route showed preference to E3.

The multivariate analysis in **Supplementary Figure 8C** showed that the effect of these factors weakened with the growth of infants and thus they may only have a short-term impact on the gut microbiome.

4. On page 10, did the pathway analyses correct for multiple testing?
Response: Yes, multiple testing correction has been conducted in the pathway analyses (Page 10, Line 33).

5. Figure S9 needs to add individual data points.
Response: As suggested, the individual data points have been added in Figure S9 (now Supplementary Figure 11).

Second round of review

Reviewer 1

I like how the authors addressed the issue of cluster count across years with subsampling (new suppl figure 2) and find the results quite convincing. However, I still think Suppl. Figure 1, which remains unchanged, will be very confusing for readers. Text must be added to explain why there are 3 clusters here and 4 later. This should include a comment about where the Prevotella samples end up in this 3-cluster scheme. Are they concentrated in one cluster or distributed across multiple clusters? This would be useful for understanding how the approach is improved by looking within each year. Suppl fig. 1 is actually a demonstration of why simply clustering all data together would be misleading and lead to missing out on seeing the Prevotella cluster. This must be explained in the text.

I am glad to hear that the enterotypes are seen with both 16S and metagenomic data. I think the authors' approach to answering this question is reasonable. However, it appears that approx. 40% of E4 metagenomic samples are classified differently depending on whether they are clustered with only metagenomic data or along with 16S data. This seems high. What is going on here?

Thank you for the clarification on sample sizes. Please add N numbers to the figure legends everywhere. If panels within one figure have different N values, this should be specified for each.

The section entitled "Clinical factors influence the prevalence rather than the transition of enterotypes" needs to be re-thought. If multivariate analyses cannot be done, then no influences can be established, only correlations/associations. The possible confounding of correlates must be tested or acknowledged plainly. For example, E1 is associated with lower gestational age and with c-section (Figure 4). But preterm infants are more often delivered by c-section, so this could explain the c-section enrichment in E1. Without checking the confounds, there's no way to say whether E1 is associated with c-section or rather with gestational age. If this is clarified in suppl. Fig 8. Then this data should replace the data in the main figure. Another example: though Fig 4B

and 4C show that E1 is greatly enriched in preterm infants across time, the PERMANOVA results indicate that this is only statistically significant in the first month of life. At the very least, the PERMANOVA results should be added to the main figure and text added to describe the possible confounds.

Also, figure 1 showed geographic associations. These could easily be caused by differences in studies. E.g. E1 seems enriched in China, is this because there were more samples collected there from preterm infants? (more studies on preterm infants)

To also address 'batch effects' the "study" variable should be added to the PERMANOVA. Or at least tested separately so the other effect sizes can be put into context. In fact, if this wasn't addressed elsewhere, it definitely should be. Different studies might have different e.g. DNA extraction techniques, storage conditions, primer choices, etc that could greatly bias the taxonomic composition. It will not be possible to get the data for all these methodological choices, but the effect of study (batch effect) should be assessed and discussed.

I am very glad to see the metadata, genus level profiles, and scripts shared. Excellent. However, a few issues need to be addressed:

Please clarify what the following column headers mean (e.g. add to the readme):

- what does "type" mean? What is 1 and 2?
- what does "food" mean? What are the numbers?

Ideally, every column should be explained, but the others are more obvious.

Also provide a key for the study terms and the study info should be provided. E.g. "preterm4" is not usable by others to link to the original study.

Why do some samples file have IDs like: "Nat0034mec" and "Six0003Bamec"? these should be all be SRR ids (in both metadata files)

While I appreciate the authors' response, the TEDDY study and analysis needs more discussion. It has nearly the same size as this paper's dataset (10k samples and 900 infants) and the TEDDY authors analyse their data as clusters and describe the transition between clusters across time. These are three key points in this paper, so I think more discussion is warranted, at least to make the reader aware that there are other views on similar data.

For example, the authors should mention explicitly that Stewart et al. did create clusters (10 clusters with varying enrichment across time) and did map the transitions of infants across time through these clusters. If there is a difference between "enterotypes" and "clusters" beyond semantics, the authors could describe this. To me, they seem like the same concept. I agree that the authors did a much better job of validating their clusters in this study than was done in the teddy work. Some of the TEDDY clusters appear to correspond to the authors' enterotypes (e.g.

TEDDY clusters 1 and 2 have high Bifidobacterium and so are similar to E2) and some do not (there is no Prevotella abundant – is this because of timeframe in life?). Disagreements between the findings of two papers is of course fine, but it is much more useful to the reader if it is described explicitly.

For future reference for the authors, the TEDDY data is available under NCBI dbGaP.

Reviewer 2

Authors have sufficiently addressed my comments. However, there are still a couple of minor issues.

1. The impact of DNA isolation methods on enterotype.
Authors has compared the enterotypes of 16S and metagenomics data separately. However, it is known that different DNA isolation methods can seriously bias the microbial composition and enterotype clustering. This question remains to be answered
2. The stratification of enterotypes driven by developmental stage rather than geographic environment
The argument from author can be true, however it is still specious as there is no meta-data to directly support the conclusion. I suggest authors to downplay the conclusion and make an appropriate statement on their data and possible underlying causes.

Dear Kevin,

We greatly appreciate your consideration of our manuscript and providing us an opportunity to address the concerns and comments raised by the reviewers. We agree with the changes you have made in the abstract of our manuscript. We also thank the reviewers for their thoughtful commentary and recommendations for improvement. All of these comments have been seriously considered and improvements have been made in this revised manuscript.

A new version of source code in the analyses of our manuscript has been deposited in ZENODO with doi: <http://doi.org/10.5281/zenodo.4756091>. A reference list of datasets included in our study has been added as an additional file (**Additional file 2**) and all additional files were renamed as required.

Appended to this letter is our point-by-point responses to the comments raised by the reviewers. The comments are reproduced and our responses are given directly afterward in a different color (blue).

Best wishes,

Fangqing Zhao

Computational Genomics Lab,
Beijing Institutes of Life Science, Chinese Academy of Sciences
Beijing CHINA

Reviewer reports:

Reviewer #1 (Remarks to the Author):

I like how the authors addressed the issue of cluster count across years with subsampling (new suppl figure 2) and find the results quite convincing. However, I still think Suppl. Figure 1, which remains unchanged, will be very confusing for readers. Text must be added to explain why there are 3 clusters here and 4 later. This should include a comment about where the *Prevotella* samples end up in this 3-cluster scheme. Are they concentrated in one cluster or distributed across multiple clusters? This would be useful for understanding how the approach is improved by looking within each year. Suppl fig. 1 is actually a demonstration of why simply clustering all data together would be misleading and lead to missing out on seeing the *Prevotella* cluster. This must be explained in the text.

Response: We thank the reviewer for this kind comment. As suggested, we have removed the previous Suppl. Figure 1 to avoid misleading and clarified this point in Discussion (Page 12 Line 31-35, Page 13 Line 1-4):

“Due to its low prevalence (13% in the second year and 6.2% the third year) in our study, the *Prevotella* enterotype was overshadowed by the other three more abundant

enterotypes when using the full dataset, in which most samples that should have been classified into the *Prevotella* enterotype (72.9%) were falsely assigned to E3 (the *Bacteroides* enterotype). However, if we clustered enterotypes by sampling the same number of samples from each year, four distinct enterotypes were clearly observed (**Additional file1: Fig. S1**), regardless of the sampling size selected. This clustering result is reproducible when using different clustering methods or even metagenomic datasets. These results emphasize the importance of sampling balance for enterotype clustering.”

I am glad to hear that the enterotypes are seen with both 16S and metagenomic data. I think the authors’ approach to answering this question is reasonable. However, it appears that approx. 40% of E4 metagenomic samples are classified differently depending on whether they are clustered with only metagenomic data or along with 16S data. This seems high. What is going on here?

Response: We greatly appreciate the reviewer’s comments. As suggested, we investigated the reason for the divergence between the results of metagenomic data and 16S data, and found that it’s also caused by the sampling imbalance. Considering that the metagenomic data almost came from the first two years of life (**Additional file1: Fig. S2B**), the proportion of E4 samples in the metagenomic data was much lower than that in 16S data (1.9% vs. 7.3%). Therefore, as discussed above, E4 is easily overshadowed by the other three more abundant enterotypes in the metagenomic data. However, the conclusion of enterotype clustering in our study is still convincing as mentioned in our previous responses (**Additional file1: Fig. S1**).

Thank you for the clarification on sample sizes. Please add N numbers to the figure legends everywhere. If panels within one figure have different N values, this should be specified for each.

Response: As suggested, the number of samples has been added to the figure legends.

The section entitled “Clinical factors influence the prevalence rather than the transition of enterotypes” needs to be re-thought. If multivariate analyses cannot be done, then no influences can be established, only correlations/associations. The possible confounding of correlates must be tested or acknowledged plainly. For example, E1 is associated with lower gestational age and with c-section (Figure 4). But preterm infants are more often delivered by c-section, so this could explain the c-section enrichment in E1. Without checking the confounds, there’s no way to say whether E1 is associated with c-section or rather with gestational age. If this is clarified in suppl. Fig 8. Then this data should replace the data in the main figure. Another example: though Fig 4B and 4C show that E1 is greatly enriched in preterm infants across time, the PERMANOVA results indicate that this is only statistically significant in the first month of life. At the very least, the PERMANOVA results should be added to the main figure and text added to describe the possible confounds.

Response: Not all the public datasets used in our study provide sufficient clinical information. For example, 6,494 samples had information of delivery mode; 6,139

samples had information of breastfeeding options; 2,986 samples had information of gestation age; while only 2,287 samples provided all these kinds of information. Considering that PERMANOVA analysis on this sub-dataset (n = 2,287) only explained part of confounds, after performing PREMANOVA analysis on multiple clinical factors, we then implemented statistical test on different clinical factors separately to explore the prevalence of four enterotypes. As suggested, we have clarified this point in the main text (Page 9 Line 6-15 and Page 13 Line 10-12) and added the PREMANOVA results to the main figure (Fig. 4A).

Also, figure 1 showed geographic associations. These could easily be caused by differences in studies. E.g. E1 seems enriched in China, is this because there were more samples collected there from preterm infants? (more studies on preterm infants)

Response: Yes, the proportion of preterm infants in the China dataset is 42%, and that in the USA is 72.9%. To avoid the bias of preterm studies, we re-clustered enterotypes based on full-term datasets (Re. Fig. 1, n = 11,909) and found that E1 and E2 were still prevalent in younger ages and developing countries. The consistency of enterotype transition (E1/E2 → E3/E4) in all countries confirms that enterotypes are associated with developmental stages rather than differences in studies.

Re. Fig. 1. Geography-related pattern of enterotypes based on full-term datasets (n = 11,909).

To also address ‘batch effects’ the “study” variable should be added to the PERMANOVA. Or at least tested separately so the other effect sizes can be put into context. In fact, if this wasn’t addressed elsewhere, it definitely should be. Different studies might have different e.g. DNA extraction techniques, storage conditions, primer choices, etc that could greatly bias the taxonomic composition. It will not be possible to get the data for all these methodological choices, but the effect of study (batch effect) should be assessed and discussed.

Response: We understand the reviewer’s concerns. Actually, batch effect is a common problem in meta-analyses. At the beginning of our research, we have tried to remove batch effects using a traditional batch-correction method (Combat). To further explore the potential batch effects in these datasets, we added more analyses and the related discussion in the main text (Page 12 Line 3-6):

1. First of all, none study-specific enterotype was observed and most studies contained over two enterotypes (**Re. Fig. 2A**). This indicates that the enterotype clustering is independent of studies. In addition, the composition of enterotypes was highly correlated with the development of infants. E1 was enriched in all preterm studies due to its feature of immaturity, which was, obviously, not derived from the bias of batch effects (**Re. Fig. 2A**). Another example is that, in one study (term13), in which all infants were collected from the first month, E1 was prevalent; while in another study (preterm4), the existence of E3 and E4 was observed because some of samples in this study were collected from infants of the second year (**Re. Fig. 2A**). These observations confirm that enterotypes are related to developmental stages and the divergence of different studies exerts slight influence on enterotype clustering.
2. As suggested, we collected DNA isolation information from the public studies included in our study and divided them into four groups to evaluate the potential influence of different DNA extraction methods (QIAGEN, MoBio, Omega and other companies). As shown in **Re. Fig. 2A**, none isolation-dependent pattern was observed, where composition of enterotypes may vary even in the same isolation method group, and similar pattern may be observed between different groups. We then clustered enterotypes using random sampling from 3 years based on different DNA isolation methods. As shown in **Re. Fig. 2B**, four enterotypes were observed, and each group clustered into three enterotypes, indicating that the enterotype clustering is independent of studies and the influence of batch effect is very limited.

Re. Fig. 2. The effect of “study” on the enterotype clustering analysis. (A) The composition of enterotypes (left) and the composition of developmental stages (right) in each study. The grey bar on the left of the panel represents different products of DNA isolation method. (B) Enterotype clustering on fecal samples processed by different DNA isolation kits (QIAGEN, MoBio). Note: there are no 3rd year samples in the MoBio group.

I am very glad to see the metadata, genus level profiles, and scripts shared. Excellent. However, a few issues need to be addressed:

Please clarify what the following column headers mean (e.g. add to the readme):

- what does “type” mean? What is 1 and 2?
- what does “food” mean? What are the numbers?

Ideally, every column should be explained, but the others are more obvious.

Response: Sorry for the unclear description of metadata. The “type” column represents the enterotype of each sample, and the “food” column refers to the duration of breastfeeding of each subject. We have modified this in the metadata and have added the explanation in the README file (<http://doi.org/10.5281/zenodo.4756091>).

Also provide a key for the study terms and the study info should be provided. E.g. “preterm4” is not usable by others to link to the original study.

Response: As suggested, a detail reference list has been added in the additional file (**Additional file 2**).

Why do some samples file have IDs like: “Nat0034mec” and “Six0003Bamec”? these should be all be SRR ids (in both metadata files)

Response: As suggested, we have corrected our sample IDs to SRR IDs.

While I appreciate the authors’ response, the TEDDY study and analysis needs more discussion. It has nearly the same size as this paper’s dataset (10k samples and 900 infants) and the TEDDY authors analyses their data as clusters and describe the transition between clusters across time. These are three key points in this paper, so I think more discussion is warranted, at least to make the reader aware that there are other views on similar data.

For example, the authors should mention explicitly that Stewart et al. did create clusters (10 clusters with varying enrichment across time) and did map the transitions of infants across time through these clusters. If there is a difference between “enterotypes” and “clusters” beyond semantics, the authors could describe this. To me, they seem like the same concept. I agree that the authors did a much better job of validating their clusters in this study than was done in the teddy work. Some of the TEDDY clusters appear to correspond to the authors’ enterotypes (e.g. TEDDY clusters 1 and 2 have high Bifidobacterium and so are similar to E2) and some do not (there is no Prevotella abundant – is this because of timeframe in life?). Disagreements between the findings of two papers is of course fine, but it is much more useful to the reader if it is described explicitly.

Response: Some similarities were observed between our study and TEDDY study; however, the biggest difference is that we divided the early development of gut microbiota into two phases (immaturity and maturity phase) while TEDDY divided them into three (developmental, transition, and stable phase). Owing to obvious distinctions among four enterotypes, the transition of two phases in our study exhibited significant difference. For example, E1 and E2 are enriched in immaturity phase, where E1 is more prevalent in preterm infants, characterizing with the dominance of Firmicutes phyla and *Bifidobacterium*. E3 and E4 are enriched in the maturity phase, in which *Bacteroides* and *Prevotella* have apparent advantages. In contrast, although 10 clusters were identified in the TEDDY study, they were difficult to distinguish from each other, which is not intuitive to describe and quantify gut microbiota transition with infant development. As suggested, we have made a comparison between the TEDDY study and ours in the following table and added more discussion in our manuscript (Page 12 Line 7-24).

Re. Table 1. Similarities and differences between TEDDY study and this study

	Our study	TEDDY study	
Similarities	Collecting over 10 thousand samples		
	Including first 3 years of life		
	Covering multiple countries		
	Bifidobacterium dominant on the early stage		
	Bacteroides associated with maturation		
Countries included	17 countries	4 countries	
Clustering method	PAM	DMM	
Clusters or enterotypes	4 enterotypes	10 clusters	
Differences	Developmental stages	2 phases	3 phases

Although using different methods, both the “cluster” in the TEDDY study and the “enterotype” in our study are efficient to simplify the microbial community analysis. The cluster 1 and 9 in the TEDDY study is corresponding to our E2 and E3, respectively. However, E1 and E4, which represents an earlier stage and a later stage in the development of infants, respectively, were not observed in TEDDY study.

For future reference for the authors, the TEDDY data is available under NCBI dbGaP.
Response: We thank the reviewer for telling us the availability of the TEDDY data.

Reviewer #2 (Remarks to the Author):

Authors have sufficiently addressed my comments. However, there are still a couple of minor issues.

1. The impact of DNA isolation methods on enterotype.

Authors has compared the enterotypes of 16S and metagenomics data separately. However, it is known that different DNA isolation methods can seriously bias the microbial composition and enterotype clustering. This question remains to be answered
Response: We thank the reviewer for pointing this out. As suggested, we added more analyses on the impact of DNA isolation methods on enterotype. Please refer to our responses to the first reviewer.

2. The stratification of enterotypes driven by developmental stage rather than geographic environment

The argument from author can be true, however it is still specious as there is no meta-data to directly support the conclusion. I suggest authors to downplay the conclusion and make an appropriate statement on their data and possible underlying causes.

Response: We greatly appreciate the reviewer’s comments. As suggested, we have modified the statement and added discussion in the main text (Page 6 Line 27-29, Page 7 Line 3, and Page 11 Line 19-21).

Third round of review

Reviewer 1

I thank the authors for their detailed handling of the points in my review. I make a few additional points below.

Section on clinical associations / Figure 4:

Based on the PERMANOVA analysis, the effect of breastmilk is effectively absent when considered in combination with the other factors. Therefore, it cannot be argued that there is a direct association between breastmilk and enterotype based on the data available. The wording of that section must be edited to warn the reader that the relationships shown in Fig 2B-E are only descriptive and may likely represent indirect effects (e.g. maybe preterm infants are more often fed with breastmilk for longer, so the association between E1 and breastfeeding duration could simply be a by-product). This is particularly true for breastfeeding where the independent explanatory power is close to zero. The low effect of delivery mode in the PERMANOVA may also indicate that c-section (which is typical for preterm infants) is also not a driving factor but just a side-effect of the relationship with prematurity.

As mentioned by Reviewer 2 as well, the possible confounding between age and geography is not yet adequately handled. A PERMANOVA analysis should also be added for relationships between enterotypes and geography and age as a first part of the section “Distinct enterotypes correspond to different developmental stages of the infant gut microbiota” (Figure 2). Geography should also be added to the PERMANOVA in Figure 4.

The key metadata variables (at least: age, geography, preterm status, birth mode) must be analysed for collinearity / checked for confounding. The results from this must be stated explicitly. This will provide context and possible explanations for why univariate associations are seen (e.g. with breast milk) but are not supported when a better multivariate analysis is performed.

I appreciate all the clarifications the authors provided for the data on Zenodo and applaud their contribution to open, reproducible science. Please also add Additional File 2 to zenodo.

Abstract:

The sentence: “Clinical information was integrated to understand outcomes of different developmental patterns.” Should be edited to add the number of samples for which clinical data is available (2,287), since this is significantly lower than the total (which is understandable, but still needs stating).

Re this sentence: “As shown in Fig. 2A, the differences in gut microbiota were much greater among enterotypes than among countries, regardless of the infant age.” I do not agree that this is shown explicitly in the figure. For example, it looks like E4 is dominated only by USA (grey), whereas samples from Luxembourg are only seen in E1 and E3? Such a statement must be tested statistically.

Authors Response

Point-by-point responses to the reviewers’ comments:

Reviewer #1:

I thank the authors for their detailed handling of the points in my review. I make a few additional points below.

Section on clinical associations / Figure 4:

Based on the PERMANOVA analysis, the effect of breastmilk is effectively absent when considered in combination with the other factors. Therefore, it cannot be argued that there is a direct association between breastmilk and enterotype based on the data available. The wording of that section must be edited to warn the reader that the relationships shown in Fig 2B-E are only descriptive and may likely represent indirect effects (e.g. maybe preterm infants are more often fed with breastmilk for longer, so the association between E1 and breastfeeding duration could simply be a by-product). This is particularly true for breastfeeding where the independent explanatory power is close to zero. The low effect of delivery mode in the PERMANOVA may also indicate that c-section (which is typical for preterm infants) is also not a driving factor but just a side-effect of the relationship with prematurity.

Response: We thank the reviewer for this comment. As suggested, we have revised our manuscript and clarified this point in Discussion (Page 9, Line 15-20, Line 33; Page 13, Line 17-27).

As mentioned by Reviewer 2 as well, the possible confounding between age and geography is not yet adequately handled. A PERMANOVA analysis should also be added for relationships between enterotypes and geography and age as a first part of the section “Distinct enterotypes correspond to different developmental stages of the infant gut microbiota” (Figure 2). Geography should also be added to the PERMANOVA in Figure 4.

Response: As suggested, geography and age were added to the PERMANOVA in Figure 4A.

The key metadata variables (at least: age, geography, preterm status, birth mode) must be analysed for collinearity / checked for confounding. The results from this must be stated explicitly. This will provide context and possible explanations for why univariate associations are seen (e.g. with breast milk) but are not supported when a better multivariate analysis is performed.

Response: As suggested, the key metadata variables have been analyzed in Figure 4A. We have also modified the statement and added discussion in the main text (Page 9, Line 15-20, Line 33; Page 13, Line 17-27).

I appreciate all the clarifications the authors provided for the data on Zenodo and applaud their contribution to open, reproducible science. Please also add Additional File 2 to zenodo.

Response: We appreciate the reviewer’s praise and as suggested, Additional File 2 has been added to Zenodo, which can be accessed at <https://zenodo.org/record/5141515>.

Abstract:

The sentence: “Clinical information was integrated to understand outcomes of different developmental patterns.” Should be edited to add the number of samples for which clinical data is available (2,287), since this is significantly lower than the total (which is understandable, but still needs stating).

Response: As suggested, we have added the sample number in Abstract (Page 2, Line 10).

Re this sentence: “As shown in Fig. 2A, the differences in gut microbiota were much greater among enterotypes than among countries, regardless of the infant age.” I do not agree that this is shown explicitly in the figure. For example, it looks like E4 is dominated only by USA (grey), whereas samples from Luxembourg are only seen in E1 and E3? Such a statement must be tested statistically.

Response: We thank the reviewer for pointing this out. As suggested, a statistical test has been added in our manuscript (Page 6, Line 5-6).